# Initial mean arterial blood pressure (MABP) measurement is a risk factor for mortality in hypertensive COVID-19 positive hospitalized patients

Tenzin Yangchen[1], Farrukh M. Koraishy[1,2], Chang Xu[3,4], Wei Hou[5], Rajeev Rohatgi[3,4]*

1 Stony Brook University School of Medicine, Stony Brook, NY, United States of America, 2 Northport VAMC, Northport, NY, United States of America, 3 James J. Peters VAMC, Bronx, NY, United States of America, 4 Icahn School of Medicine at Mount Sinai, One Gustave L. Levy Place, New York, NY, United States of America, 5 Vertex Pharmaceuticals Inc., Boston, MA, United States of America

* Rajeev.Rohatgi@va.gov

## Abstract

### Background

Hypertension (HTN) is associated with severe COVID-19 infection; however, it remains unknown if the level of blood pressure (BP) predicts mortality. We tested whether the initial BP in the emergency department of hospitalized patients portends mortality in COVID-19 positive(+) patients.

### Methods

Data from COVID-19(+) and negative (-) hospitalized patients at Stony Brook University Hospital from March to July 2020 were included. The initial mean arterial BPs (MABPs) were categorized into tertiles (T) of MABP (65–85 [T1], 86–97 [T2] and ≥98 [T3] mmHg). Differences were evaluated using univariable (t-tests, chi-squared) tests. Multivariable (MV) logistic regression analyses were computed to assess links between MABP and mortality in hypertensive COVID-19 patients.

### Results

1549 adults were diagnosed with COVID-19 (+) and 2577 tested negative (-). Mortality of COVID-19(+) was 4.4-fold greater than COVID-19(-) patients. Though HTN prevalance did not differ between COVID-19 groups, the presenting systolic BP, diastolic BP, and MABP were lower in the COVID-19(+) vs (-) cohort. When subjects were categorized into tertiles of MABP, T2 tertile of MABP had the lowest mortality and the T1 tertile of MABP had greatest mortality compared to T2; however, no difference in mortality was noted across tertiles of MABP in COVID-19 (-). MV analysis of COVID-19 (+) subjects exposed death as a risk factor for T1 MABP. Next, the mortality of those with a historic diagnosis of hypertension or normotension were studied. On MV analysis, T1 MABP, gender, age, and first respiratory rate

**Data Availability Statement:** All relevant data for this study are publicly available from the Harvard Dataverse repository (https://doi.org/10.7910/DVN/YG01YI).

**Funding:** RR 1I01BX003015 VA Merit Review
https://www.research.va.gov/services/shared_
docs/resources.cfm The funders had no role in
study design, data collection and analysis, decision
to publish, or preparation of the manuscript.

**Competing interests:** The authors have declared
that no competing interests exist.

correlated with mortality while lymphocyte count inversely correlated with death in hypertensive COVID-19 (+) patients while neither T1 nor T3 categories of MABP predicted death in non-hypertensives.

## Conclusions

Low-normal admitting MABP in COVID-19 (+) subjects with a historical diagnosis of HTN is associated with mortality and may assist in identifying those at greatest mortality risk.

## Introduction

The COVID-19 pandemic, to date, led to more than 900,000 deaths in the United States, with the Northeast United States exposed to high mortality rate during the initial wave (March-July 2020) of the infection. SARS-CoV2, the etiologic agent of COVID-19 illness, infects epithelia of the upper airways, lungs, kidney and gastrointestinal tract. To adhere to epithelia, the spike protein of SARS-CoV2 is cleaved by serine proteases to bind the angiotensin converting enzyme 2 (ACE2) receptor and enters the cell for replication [1]. ACE2 is a metalloprotease which cleaves Angiotensin (Ang) I to Ang 1–9 and Ang II to Ang 1–7 [2]. In the renin-angiotensin-aldosterone system (RAAS), ACE2 activity serves to antagonize the hypertensive and vasoconstrictive effects of Ang II by reducing Ang II expression and, indirectly, by enhancing Ang 1–7 content to bind the Mas receptor [2]. When SARS-CoV2 binds the ACE2 receptor it inhibits ACE2 enzymatic activity and reduces tissue expression of ACE2, which theoretically leads to disproportionate vasoconstrictive Ang II effects. Serine proteases, which cleave SARS-CoV2 spike protein, also cleave and activate the renal epithelial Na channel to absorb Na in the renal collecting duct and mediate hypertension, and thereby may explain the excess prevalence and morbidity in hypertensive COVID-19 positive (+) patients [3, 4]. Moreover, cross-sectional human data suggests comorbidities, including hypertension lead to greater ACE2 expression in lungs [5, 6]. Therefore, systemic hypertension (HTN) is hypothesized as a risk factor for severe COVID-19 infection leading to mortality.

Several observational, cross-sectional investigations confirm an association between chronic systemic HTN and severe COVID-19 illness and mortality; however, the association of presenting systemic blood pressure (BP) level and mortality in patients with COVID-19 has not been reported [7–11]. Due to SARS-CoV-2's direct effects on the RAAS system, we hypothesized that BP level at clinical presentation would be a potent risk factor for mortality, differing from uninfected patients. To this end, a retrospective analysis was performed on a database of patients admitted to the Stony Brook University Hospital (SBUH) from March 2020 to July 2020 which included COVID-19 (+) and negative (-) patients.

## Methods

### Participants

Historical, clinical and laboratory data were abstracted from patients admitted to SBUH from March 2020 to July 2020 during the first wave of the COVID-19 pandemic. Polymerase chain reaction (PCR) of nasopharyngeal swabs identified those with COVID-19. Only the first wave of the pandemic was analyzed because natural immunity and vaccination confound the effects of BP on mortality. 4126 subjects were admitted to SBUH during this period: 1549 diagnosed with COVID-19 and 2577 testing negative for COVID-19. Though multiple clinical and

laboratory parameters were abstracted from the electronic health records (EHR) at the time of admission for each patient, the focus of this investigation was to identify which historical, clinical and laboratory parameters predicted death. To this end, only the initial measures collected during the emergency room evaluation were included in the analysis. 91 variables were collected (See S1 Table) and included age, gender, race, ethnicity, comorbid conditions, medications, vital signs, respiratory measures, severity of disease, renal lab tests, and inflammatory markers. All comorbid conditions including HTN were abstracted from existing ICD-9/10 codes. S1 Table includes abbreviations and units for the data collected in the subsequent tables. Dialysis patients, transplant recipients, subjects less than 18 years old and pregnant women were excluded from the analysis to generate a homogenous group.

The study the was approved by the SBUH Institutional Review Board (IRB2020-00239; Characterization of AKI with outcomes in patients with COVID-19).

## BP measures

BP markers (explanatory variables) were treated as categorical variables. Since hypotension is an established predictor of death in hospitalized patients, subjects with hypotension, defined as mean arterial BP(MABP)< 65 mm Hg were excluded from further analysis. Subjects were also divided into tertiles of MABP: T1 (65–85 mm Hg), T2 (86–97 mm Hg) and T3 ($\geq$98 mm Hg). The MABP category with the lowest mortality was identified as the control group and other two groups compared to it. The T2 MABP group had the lowest mortality amongst the COVID-19 (+), but the highest mortatlity amongst the COVID-19 (-) patients. T2 MABP were selected as the control groups in both COVID-19(+) and (-).

## Statistical analyses

Data analysis was performed using SAS version 9.4 (SAS Institute Inc., Cary, N.C.). Descriptive statistics (frequencies, proportions, mean standard deviation, median and interquartile interval [IQR]) were used to compare the demographics, comorbid conditions, the severity of illness, clinical, and laboratory measures between COVID-19 (+) and (-) patients as well as hypertensive and non-hypertensive subjects. Differences were evaluated using univariable (t-tests, Mann-Whitney tests, and chi-squared tests) for continuous and categorical data, respectively. Multivariable (MV) logistic regression analyses were computed to assess the associations between MABP and clinical characteristics in COVID-19 hospitalized patients and the association of mortality to MABP in hypertensive COVID-19 (+) subjects. Demographic variables and variables noted to be significant on univariate analysis were used. ACEis were also studied because of their theoretic effects on RAAS. Variables with less than 80% observations were excluded. A value of P $\leq$ 0.05 was considered statistically significant.

## Results

4126 patients were admitted to SBUH. 1549 were diagnosed with and 2577 without COVID-19. 210 (13.6%) of COVID-19 (+) died during the study period while only 81 (3.1%) of COVID-19 (-) patients expired (Table 1), corresponding to a 4.4-fold relative increase in mortality among the COVID-19 (+) group. Patients with COVID-19 were older, and more likely to be male, non-White and Hispanic compared to those without COVID-19 (Table 1). Diabetes was more prevalent in the COVID-19 (+) than COVID-19 (-) patients. The diagnosis of HTN did not differ between COVID-19 (+) (627/1549; 40.5%) vs. COVID-19 (-) (1011/2577; 39.2%) patients, however, the average presenting systolic BP (SBP), diastolic BP (DBP) and MABP was significantly lower in the COVID-19 (+) cohort (p<0.05; Table 1).

**Table 1. Comparison of hospitalized COVID-19 positive and negative patients.**

| Variables | COVID-19 Status | | p value |
|---|---|---|---|
| | COVID-19 Positive n = 1549 | COVID-19 Negative n = 2577 | |
| **Demographics** | | | |
| Age (years) | 61.09 (18.51) | 57.78 (21.44) | <0.0001* |
| **Gender** | | | |
| Female | 706 (45.6%) | 1403 (54.4%) | <0.0001* |
| Male | 843 (54.4%) | 1174 (45.6%) | |
| **Race** | | | |
| American Indian or Alaska Native | 2 (0.1%) | 0 (0.0%) | <0.0001* |
| Asian | 53 (3.4%) | 63 (2.4%) | |
| Black or African American | 107 (6.9%) | 161 (6.2%) | |
| Other Race/Unknown | 471 (30.4%) | 292 (11.3%) | |
| White | 918 (59.3%) | 2059 (79.9%) | |
| **Ethicity** | | | |
| Hispanic or Latino | 270 (17.4%) | 117 (4.5%) | <0.0001* |
| Not Hispanic | 1279 (82.6%) | 2460 (95.5%) | |
| **Comorbid conditions** | | | |
| Diabetes mellitus | 456 (29.4%) | 552 (21.4%) | <0.0001* |
| HF | 238 (15.4%) | 431 (16.7%) | 0.251 |
| CKD | 210 (13.6%) | 331 (12.8%) | 0.511 |
| COPD | 171 (11.0%) | 324 (12.6%) | 0.142 |
| HTN | 627 (40.5%) | 1011 (39.2%) | 0.428 |
| CAD | 280 (18.1%) | 571 (22.2%) | 0.002* |
| **Severity of illness** | | | |
| Length of Hospital Stay (days) | 8.00 (5.00, 13.00) | 5.00 (3.00, 8.00) | <0.0001* |
| Invasive Vent Days | 12.00 (7.00, 24.00) | 3.50 (2.00, 9.00) | <0.0001* |
| ICU Admission | 339 (21.9%) | 369 (14.3%) | <0.0001* |
| Length of ICU stay (days) | 10.00 (4.00, 21.00) | 4.00 (2.00, 8.00) | <0.0001* |
| Sepsis | 459 (29.6%) | 246 (9.5%) | <0.0001* |
| Vasopressor Indicator | 71 (4.6%) | 393 (15.3%) | <0.0001* |
| **Vitals** | | | |
| SBP (mm Hg) | 127.65 (24.17) | 134.40 (28.51) | <0.0001* |
| DBP (mm Hg) | 74.50 (14.81) | 76.97 (14.59) | <0.0001* |
| MABP (mm Hg) | 90.95 (14.60) | 95.16 (20.33) | <0.0001* |
| Heart Rate (beats/min) | 99.21 (46.64) | 96.10 (192.86) | 0.434 |
| Oral Temperature (˚C) | 37.38 (1.22) | 36.90 (1.29) | <0.0001* |
| Respiratory Rate (respirations/min) | 22.14 (8.53) | 19.04 (9.27) | <0.0001* |
| **Renal Labs** | | | |
| BUN (mg/dL) | 16.00 (10.50, 26.00) | 17.00 (12.00, 26.00) | 0.0006* |
| Creatinine (mg/dL) | 0.91 (0.71, 1.23) | 0.92 (0.71, 1.24) | 0.555 |
| sodium (meq/L) | 137.00 (134.00, 140.00) | 138.00 (135.00, 140.00) | <0.0001* |
| K (meq/L) | 4.10 (3.80, 4.50) | 4.20 (3.80, 4.50) | 0.026* |
| Cl (meq/L) | 98.00 (95.00, 102.00) | 100.00 (97.00, 103.00) | <0.0001* |
| hco3(meq/L) | 24.00 (21.00, 26.00) | 24.00 (22.00, 26.00) | 0.005* |
| Ca (mg/dL) | 9.00 (8.70, 9.40) | 9.40 (9.00, 9.80) | <0.0001* |
| Ca ionized (mg/dL) | 4.50 (4.20, 4.70) | 4.60 (4.30, 4.80) | 0.015* |
| Phosphate (mg/dL) | 3.20 (2.70, 3.70) | 3.30 (2.80, 3.90) | <0.0001* |
| Mg (mg/dL) | 2.00 (1.80, 2.20) | 2.00 (1.80, 2.10) | <0.0001* |

(*Continued*)

**Table 1.** (Continued)

| Variables | COVID-19 Status | | p value |
|---|---|---|---|
| | COVID-19 Positive n = 1549 | COVID-19 Negative n = 2577 | |
| **Inflammatory Labs** | | | |
| Ferritin (mcg/L) | 593.90 (249.80, 1145.0) | 206.20 (86.55, 511.20) | <0.0001* |
| Albumin Serum (mg/dL) | 3.70 (3.40, 4.00) | 4.00 (3.50, 4.30) | <0.0001* |
| Lymphocyte Count (K/mm$^3$) | 0.98 (0.69, 1.42) | 1.39 (0.90, 2.02) | <0.0001* |
| Procalcitonin (ng/mL) | 0.16 (0.09, 0.31) | 0.13 (0.07, 0.41) | 0.011* |
| D-Dimer (ng/mL) | 404.00 (239.00, 831.00) | 435.50 (231.00, 1077.0) | 0.175 |
| IL6 (pg/mL) | 48.35 (22.30, 97.20) | 18.40 (5.80, 61.10) | <0.0001* |
| WBC (10$^9$ cell/L) | 7.57 (5.56, 10.24) | 9.84 (7.43, 12.91) | <0.0001* |
| ESR (mm/hr) | 53.00 (29.00, 75.00) | 29.00 (14.00, 65.00) | <0.0001* |
| CRP (mg/L) | 7.35 (2.80, 14.10) | 2.50 (0.60, 7.95) | <0.0001* |
| **Other Labs** | | | |
| HB (g/dL) | 13.10 (11.60, 14.50) | 12.70 (11.20, 14.20) | <0.0001* |
| Lactate (mmol/L) | 1.50 (1.10, 2.00) | 1.60 (1.10, 2.40) | 0.001* |
| BNP (pg/mL) | 254.00 (62.00, 1189.0) | 706.00 (173.00, 2808.0) | <0.0001* |
| Troponin (ng/mL) | 0.01 (0.01, 0.01) | 0.01 (0.01, 0.02) | 0.0001* |
| INR | 1.20 (1.10, 1.30) | 1.10 (1.00, 1.30) | <0.0001* |
| LDH (U/L) | 284.00 (216.00, 374.00) | 231.00 (190.00, 305.00) | <0.0001* |
| AST (U/L) | 38.00 (25.00, 60.00) | 24.00 (18.00, 38.00) | <0.0001* |
| ALT (U/L) | 29.00 (17.00, 50.00) | 19.00 (13.00, 33.00) | <0.0001* |
| CPK (U/L) | 102.00 (51.00, 233.00) | 113.00 (55.00, 357.00) | 0.018* |
| **Lipid Profile** | | | |
| LDL (mg/dL) | 69.00 (49.00, 90.00) | 78.00 (54.00, 108.00) | <0.0001* |
| Triglyceride (mg/dL) | 125.00 (91.00, 188.00) | 103.00 (73.00, 159.00) | <0.0001* |
| HDL (mg/dL) | 32.00 (24.00, 40.00) | 44.00 (34.00, 55.00) | <0.0001* |
| **Mortality** | 210 (13.6%) | 81 (3.1%) | <0.0001* |

Data were shown with n (%) for categorical variables, mean (sd) and median (IQR) for continuous variables.

* p<0.05; P values were based on Chi-square tests, t-tests and Mann-Whitney tests.

Non-hypotensive (MABP >65) patients were stratified by tertiles of MABP. In COVID-19 (+) patients T1 MABP resulted in an average MABP of 77.5±5.4 mm Hg or an average BP of 115±18/68±11 mm Hg; T2 MABP of 91.5±3.4 mm Hg or an average BP of 127±18/74±11 mm Hg and T3 MABP of 108±10 mm Hg or an average BP of 146±24/83±14 mm Hg (Table 2). T2 tertile of MABP had the lowest mortality among COVID-19 (+) patients (56/551; 10.2%) and the T1 tertile demonstrated the greatest mortality at 16.1% than T2 (p<0.05; Table 2) (See S2 Table for unabridged variables). Mortality in T3 tertile of MABP did not differ from T2. No difference in mortality was noted across the MABP tertiles of COVID-19 (-) cohort (S3 Table). A MV regression model to identify the variables predicting the T1 tertile of MABP was studied. T1 tertile of MABP was compared to T2 MABP as the reference cohort since T2 MABP had the lowest mortality. In the T1 vs T2 analysis, death (odds ratio (OR) 1.709 [CI 1.123, 2.601],) and higher white blood cell count (OR 1.032 [CI 1.005, 1.183]) was associated with T1 MABP vs T2 at presentation (as shown in Fig 1). On the other hand, a higher first calcium (OR 0.769 [CI 0.597,0.992]), higher hemaglobin (OR 0.905 [CI 0.844, 0.971]), and use of enoxaparin (OR 0.743 [CI 0.554, 0.997]) were associated with T2 MABP (shown in Fig 1). The multivariate analysis informs us about the characteristics of those patients with T1 or T2 level of MABP at initial evaluation.

**Table 2. Comparison of MABP Tertiles in hospitalized COVID-19 positive patients.**

| Variables | COVID-19 Positive | | | | |
|---|---|---|---|---|---|
| | T1 ≥65- <86 mmHg n = 504 | T2# ≥86-<98 mmHg n = 551 | T3 ≥ 98 mmHg n = 452 | p- value[1] (T1 vs. T2) | p-value[2] (T3 vs. T2) |
| **Demographics** | | | | | |
| Age | 59.54 (19.48) | 60.04 (18.40) | 63.12 (17.22) | 0.869 | 0.017* |
| **Gender** | | | | | |
| Female | 260 (51.6%) | 251 (45.6%) | 180 (39.8%) | 0.1 | 0.136 |
| Male | 244 (48.4%) | 300 (54.4%) | 272 (60.2%) | | |
| **Race** | | | | | |
| Asian | 18 (3.6%) | 17 (3.1%) | 17 (3.8%) | 0.417 | 0.999 |
| Black or African American | 29 (5.8%) | 42 (7.6%) | 35 (7.7%) | | |
| Other Race/Unknown | 144 (28.6%) | 181 (32.8%) | 141 (31.2%) | | |
| White | 313 (62.1%) | 311 (56.4%) | 259 (57.3%) | | |
| **Ethnicity** | | | | | |
| Hispanic or Latino | 74 (14.7%) | 103 (18.7%) | 91 (20.1%) | 0.163 | 0.999 |
| Not Hispanic | 430 (85.3%) | 448 (81.3%) | 361 (79.9%) | | |
| **Comorbid conditions** | | | | | |
| Diabetes mellitus | 136 (27.0%) | 149 (27.0%) | 154 (34.1%) | 0.999 | 0.032* |
| HF | 79 (15.7%) | 66 (12.0%) | 80 (17.7%) | 0.163 | 0.021* |
| CKD | 63 (12.5%) | 64 (11.6%) | 69 (15.3%) | 0.999 | 0.18 |
| COPD | 55 (10.9%) | 58 (10.5%) | 48 (10.6%) | 0.999 | 0.999 |
| HTN | 171 (33.9%) | 222 (40.3%) | 221 (48.9%) | 0.066 | 0.013* |
| **Vitals** | | | | | |
| SBP (mm Hg) | 114.85 (18.06) | 126.73 (18.45) | 145.82 (24.11) | <0.0001* | <0.0001* |
| DBP (mm Hg) | 68.10 (10.95) | 74.13 (10.69) | 82.98 (13.72) | <0.0001* | <0.0001* |
| Pulse pressure difference (PPD) (mm Hg) | 46.77 (13.94) | 52.52 (16.04) | 62.82 (19.63) | <0.0001* | <0.0001* |
| MABP | 77.51 (5.36) | 91.52 (3.40) | 108.04 (9.96) | <0.0001* | <0.0001* |
| Respiratory Rate (respirations/min) | 22.08 (10.14) | 21.44 (6.91) | 22.99 (8.39) | 0.37 | 0.008* |
| Oral Temperature (˚C) | 37.46 (0.91) | 37.45 (0.86) | 37.23 (1.78) | 0.982 | 0.011* |
| **Medications** | | | | | |
| Enoxaparin | 323 (64.1%) | 391 (71.0%) | 250 (24.2%) | 0.121 | 0.999 |
| **Inflammatory Labs** | | | | | |
| Ferritin (mcg/L) | 530.40 (238.30, 1025.0) | 611.50 (267.40, 1227.0) | 598.90 (234.00, 1209.0) | 0.242 | 0.977 |
| Albumin Serum (mg/dL) | 3.60 (3.30, 4.00) | 3.80 (3.40, 4.10) | 3.70 (3.40, 4.00) | 0.0007* | 0.405 |
| Lymphocyte Count (K/mm³) | 0.94 (0.64, 1.32) | 0.96 (0.71, 1.42) | 1.03 (0.72, 1.50) | 0.324 | 0.315 |
| Procalcitonin (ng/mL) | 0.17 (0.09, 0.35) | 0.15 (0.09, 0.28) | 0.14 (0.09, 0.27) | 0.123 | 0.735 |
| D-Dimer (ng/mL) | 412.00 (238.00, 920.00) | 358.00 (227.00, 694.00) | 424.50 (247.00, 888.00) | 0.249 | 0.045* |
| IL6 (pg/mL) | 49.40 (22.70, 91.70) | 41.25 (21.60, 80.70) | 58.10 (22.30, 111.30) | 0.569 | 0.124 |
| WBC (10⁹ cell/L) | 7.87 (5.76, 10.67) | 7.24 (5.51, 9.88) | 7.66 (5.81, 10.04) | 0.064 | 0.162 |
| ESR (mm/hr) | 60.00 (31.00, 80.00) | 50.00 (29.00, 71.00) | 50.00 (28.00, 74.00) | 0.003* | 0.85 |
| CRP (mg/L) | 7.45 (3.00, 15.60) | 8.00 (3.30, 13.60) | 6.60 (2.10, 13.70) | 0.98 | 0.281 |
| HB (g/dL) | 12.80 (11.30, 14.10) | 13.30 (11.90, 14.60) | 13.60 (11.90, 15.00) | <0.0001* | 0.151 |
| **Mortality** | 81 (16.1%) | 56 (10.2%) | 58 (12.8%) | 0.009* | 0.37 |

#CONTROL GROUP

Data were shown with n (%) for categorical variables, mean (sd) and median (IQR) for continuous variables.

* p<0.05; P values were based on ANOVA with Dunnett's adjustment, Kruskal-Wallis test with DSCF adjustment and Chi-square test with Bonferroni adjustment for multiple comparisons.

p-value[1]-T1 vs. T2

p-value[2]-T3 vs. T2

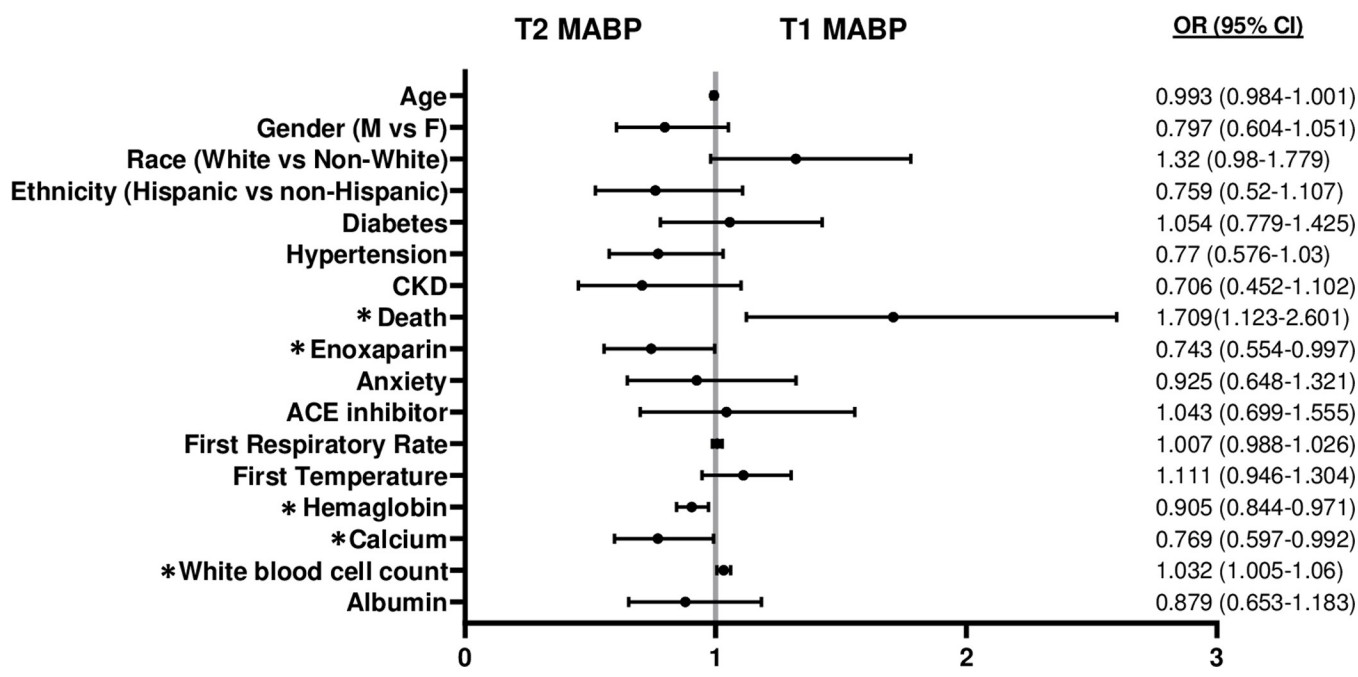

**Fig 1. The odds ratio (OR) for patient characteristics that are associated T1 compared to T2 MABP in hospitalized COVID-19 (+) patients.**

Because the diagnosis of HTN correlated with severe COVID-19 illness and death in several studies, the characteristics of those who survived and died with a pre-existing diagnosis of HTN, as well as those without a diagnosis of HTN were studied. In this cohort the mortality of hypertensive COVID-19 (+) patients (14.2%) did not differ from normotensive COVID-19 (+) patients (13.1%; p = 0.54). Next, we sought to determine whether the MABP correlated with mortality in either hypertensive and/or normotensive COVID-19 (+) subjects. COVID-19 (+) patients with a diagnosis of HTN were stratified by mortality (Table 3). Among, hypertensives T1 MABP level was more prevalent amongst those that died while T2 MABP level most more prevalent among those that survived (Table 3). An equal prevalence of T3 MABP level was found amongst the survivors and deaths (Table 3). MV logistic regression showed that T1 MABP compared to T2 MABP increased the odds of death (shown in Fig 2) by ~2-fold (OR 2.069 [CI 1.027, 4.167]) while no association was noted in T3 vs T2 MABP (shown in Fig 2). Gender (OR 1.936 M vs F [CI 1.076, 3.482] significantly contributed to the risk for death, as did Age (OR 1.037 per 5 years [CI 1.016, 1.064]) and first respiratory rate 1.074 [CI 1.035, 1.114])]) (shown in Fig 2). Greater first lymphocyte count (OR 0.502 per 1 unit increase [CI 0.284, 0.888]) portended better survival (shown in Fig 2). A similar analysis was performed in those COVID-19 (+) subjects without a diagnosis of chronic HTN (Table 4). None of MABP categories were associated with death by univariate analysis (Table 4). MV analysis of this group did not show that T1 or T3 vs T2 MABP significantly affected survival; however, gender (OR 1.776 [CI 1.026, 3.075]), increasing age (OR 1.054 per 5 years [CI 1.035, 1.074]), and first respiratory rate (OR 1.064 [CI 1.034, 1.095] contributed to the risk of death. Greater first serum albumin (OR 0.529 per 1 g/dL [CI 0.259, 0.949]), and lymphocyte count (OR 0.389 per 1 unit increase [CI 0.213, 0.71]) portended better overall survival (shown in Fig 3). Please see S5 Table for variables not present in Table 4.

**Table 3. Comparison of hypertensive COVID-19 positive patients who expired and survived.**

| Variables | Mortality Status | | |
|---|---|---|---|
| | Survival n = 538 | Non-survival n = 89 | P-value |
| **Demographics** | | | |
| Age | 66.37 (14.52) | 72.51 (13.42) | 0.0001* |
| **Gender** | | | |
| Female | 258 (48.0%) | 33 (37.1%) | 0.057 |
| Male | 280 (52.0%) | 56 (62.9%) | |
| **Race** | | | |
| Asian | 21 (3.9%) | 4 (4.5%) | 0.968 |
| Black or African American | 49 (9.1%) | 7 (7.9%) | |
| Other Race/Unknown | 132 (24.5%) | 21 (23.6%) | |
| White | 336 (62.5%) | 57 (64.0%) | |
| **Ethnicity** | | | |
| Hispanic or Latino | 65 (12.1%) | 14 (15.7%) | 0.337 |
| Not Hispanic | 473 (87.9%) | 75 (84.3%) | |
| **Comorbid conditions** | | | |
| Diabetes mellitus | 209 (38.8%) | 35 (39.3%) | 0.932 |
| HF | 42 (7.8%) | 10 (11.2%) | 0.277 |
| CKD | 45 (8.4%) | 9 (10.1%) | 0.586 |
| COPD | 61 (11.3%) | 14 (15.7%) | 0.237 |
| CAD | 538 (100.0%) | 89 (100.0%) | 0.237 |
| Anxiety | 107 (19.9%) | 24 (27.0%) | 0.128 |
| **Vitals** | | | |
| MABP T1 (65–85) | 139 (26.2%) | 32 (38.1%) | 0.042* |
| MABP T2 (86–97) | 200 (37.7%) | 22 (26.2%) | |
| MABP T3 (>98) | 191 (36.0%) | 30 (35.7%) | |
| Heart Rate (beats/min) | 96.11 (43.51) | 100.19 (24.63) | 0.206 |
| Oral Temperature (˚C) | 37.28 (1.67) | 37.39 (0.83) | 0.351 |
| Respiratory Rate (respirations/min) | 21.34 (5.97) | 25.83 (10.44) | 0.0001* |
| **Respiratory Measures** | | | |
| Pulse Ox (%) | 94.00 (91.50, 95.00) | 93.00 (88.00, 95.00) | 0.038* |
| **Renal Function** | | | |
| BUN (blood urea nitrogen, mg/dL) | 17.00 (12.00, 25.00) | 23.00 (17.00, 33.00) | <0.0001* |
| Creatinine (serum creatinine, mg/dL) | 0.93 (0.74, 1.20) | 1.09 (0.87, 1.54) | 0.0002* |
| K (serum potassium, meq/L) | 4.10 (3.80, 4.50) | 4.30 (3.95, 4.70) | 0.011* |
| hco3 (serum bicarbonate, meq/L) | 24.00 (22.00, 26.00) | 22.00 (20.00, 25.00) | 0.0002* |
| Ca (serum calcium, mg/dL) | 9.05 (8.70, 9.50) | 8.90 (8.60, 9.20) | 0.007* |
| **Inflammatory Labs** | | | |
| Ferritin (mcg/L) | 566.60 (259.80, 1184.0) | 839.40 (493.70, 1394.0) | 0.001* |
| Albumin Serum (mg/dL) | 3.70 (3.40, 4.00) | 3.40 (3.20, 3.80) | <0.0001* |
| Lymphocyte Count (K/mm$^3$) | 1.00 (0.72, 1.44) | 0.75 (0.51, 1.05) | <0.0001* |
| D-Dimer | 364.00 (225.00, 709.00) | 586.00 (349.50, 1359.0) | <0.0001* |
| WBC | 7.19 (5.33, 9.21) | 8.70 (6.21, 11.95) | 0.002* |

Data were shown with n (%) for categorical variables, mean (sd) and median (IQR) for continuous variables.

* p<0.05; P values were based on Chi-square tests, t-tests and Mann-Whitney tests.

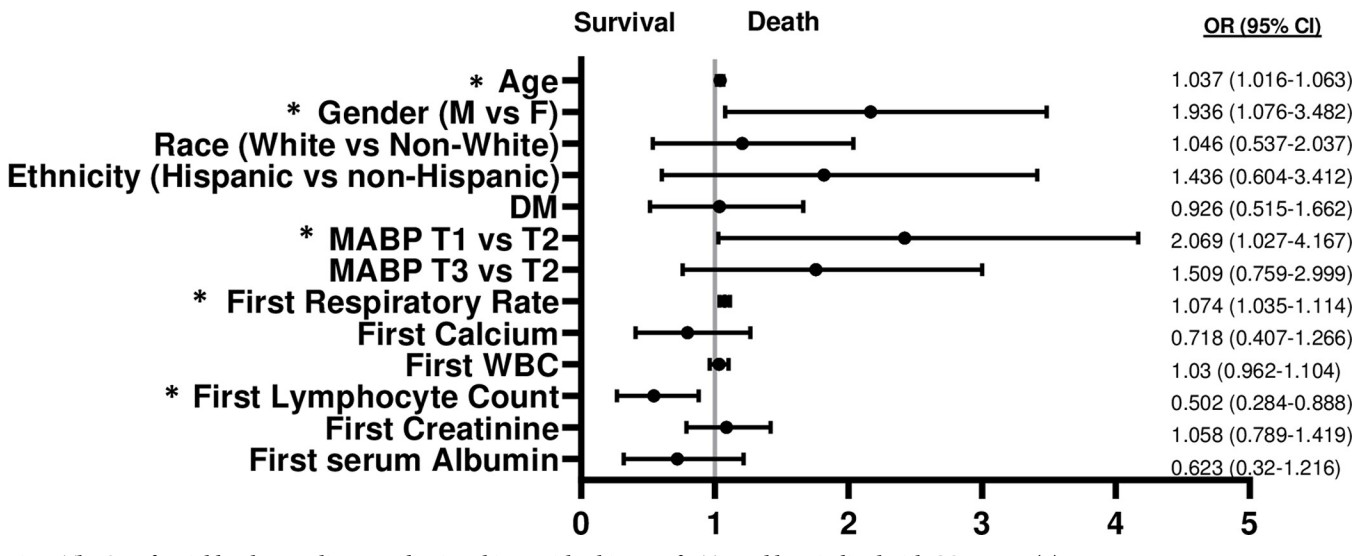

**Fig 2. The OR of variables that predict mortality in subjects with a history of HTN and hospitalized with COVID-19 (+).**

## Discussion

In conclusion, mortality is associated with low-normal BP (T1 MABP) at presentation in COVID-19 (+) patients, and a T1 MABP is risk factor for death in hypertensive COVID-19 (+) subjects. Its uncertain why low-normal MABP compared to normal MABP is associated with mortality specifically in patients with COVID-19 and chronic systemic HTN; however, clinical and basic investigations allow us to speculate why this finding is true. Clinical studies demonstrate an association between the diagnosis of chronic systemic HTN and mortality in COVID-19 (+) patients [7–12]. The renin-angiotensin system, and specifically, ACE2 receptor, the receptor for COVID-19, is increased in bronchial biopsy specimens of patients with chronic systemic HTN [5]. Thus, with greater accessible ACE2 receptor in hypertensive subjects, COVID-19 binding, entry and replication is expected to be enhanced. In most disease states, a high viral load tends to create a septic clincial picture associated with low blood pressure; however, ACE2 receptor is a negative regulator of the RAAS system and its activation reduces blood pressure. Therefore, it had been proposed that COVID-19 binding to and subsequent internalization of ACE2 receptor contributes to unopposed Ang-2 activity to raise systemic BP. In short, we tested whether presenting BP would be high (unopposed angiontensin-2 acitivty) or low (a classic septic picture, with high viral load), and whether either condition correlates to mortality. Those with abject hypotension were excluded because hypotension (MABP<65 mm Hg) is a poor prognostic sign and contributes to end-organ injury such as acute kidney injury and cardiac ischemia. Interestingly, in a univariate analysis, T1 (MABP 65–86 mm Hg; mean BP 115/68 mm Hg) level of MABP was associated with a greater mortality than T2 MABP (MABP 86–98 mm Hg; mean BP 127/74 mm Hg), even though the patients were not overtly hypotensive. In a multivariate analysis, death and white blood cell count were characteristics associated with T1 MABP level, implying an association between death and presenting T1 MABP. Why was death associated with the lowest tertile of MABP, even though the MABP is normal? The higher white blood cell count in the MV analysis suggests greater inflammation. The data abstracted cannot directly answer this question; however, long standing chronic systemic HTN contributes to microvascular disease and end-organ damage, necessitating higher systemic blood pressure to perfuse tissues. In this case, we speculate a low-normal MABP may reduce tissue perfusion in those with systemic HTN. This was confirmed,

**Table 4. Comparison of non-hypertensive COVID-19 positive patients who expired and survived.**

| Variables | Mortality Status | | |
|---|---|---|---|
| | Survival n = 801 | Non-survival n = 121 | P-value |
| **Demographics** | | | |
| Age | 54.52 (19.16) | 72.69 (15.94) | <0.0001* |
| **Gender** | | | 0.002* |
| Female | 376 (46.9%) | 39 (32.2%) | |
| Male | 425 (53.1%) | 82 (67.8%) | |
| **Race** | | | 0.002* |
| Asian | 23 (2.9%) | 5 (4.1%) | |
| Black or African American | 46 (5.7%) | 5 (4.1%) | |
| Other Race/Unknown | 294 (36.7%) | 24 (19.8%) | |
| White | 438 (54.7%) | 87 (71.9%) | |
| **Ethicity** | | | 0.008* |
| Hispanic or Latino | 177 (22.1%) | 14 (11.6%) | |
| Not Hispanic | 624 (77.9%) | 107 (88.4%) | |
| **Comorbid conditions** | | | |
| Diabetes mellitus | 176 (22.0%) | 36 (29.8%) | 0.058 |
| HF | 142 (17.7%) | 44 (36.4%) | <0.0001* |
| CKD | 103 (12.9%) | 53 (43.8%) | <0.0001* |
| COPD | 67 (8.4%) | 29 (24.0%) | <0.0001* |
| CAD | 103 (12.9%) | 46 (38.0%) | <0.0001* |
| Anxiety | 111 (13.9%) | 13 (10.7%) | 0.349 |
| **Vitals** | | | |
| MABP T1 (65–85) | 284 (36.3%) | 49 (44.1%) | 0.229 |
| MABP T2 (86–97) | 295 (37.7%) | 34 (30.6%) | |
| MABP T3 (>98) | 203 (26.0%) | 28 (25.2%) | |
| Heart Rate (beats/min) | 100.49 (47.29) | 103.89 (64.63) | 0.580 |
| Oral Temperature (˚C) | 37.47 (0.90) | 37.20 (0.81) | 0.002* |
| Respiratory Rate (respirations/min) | 21.54 (8.96) | 27.03 (11.14) | <0.0001* |
| **Respiratory Measures** | | | |
| Pulse Ox | 95.00 (93.00, 98.00) | 92.00 (86.00, 96.00) | <0.0001* |
| PaO2 | 81.00 (68.00, 113.00) | 78.00 (62.00, 109.00) | 0.281 |
| FiO2 | 50.00 (40.00, 50.00) | 50.00 (50.00, 100.00) | 0.0001* |
| **Renal Function** | | | |
| BUN | 13.00 (9.00, 21.00) | 28.00 (18.00, 45.00) | <0.0001* |
| creatinine | 0.85 (0.66, 1.10) | 1.38 (0.98, 1.78) | <0.0001* |
| sodium | 137.00 (134.00, 140.00) | 138.00 (133.00, 143.00) | 0.014* |
| K | 4.10 (3.80, 4.50) | 4.30 (4.00, 4.80) | <0.0001* |
| Cl | 99.00 (95.00, 102.00) | 99.00 (95.00, 103.00) | 0.312 |
| hco3 | 24.00 (21.00, 26.00) | 22.00 (20.00, 25.00) | 0.003* |
| Ca | 9.00 (8.70, 9.40) | 9.00 (8.60, 9.30) | 0.185 |
| Ca ionized | 4.50 (4.30, 4.70) | 4.40 (4.10, 4.60) | 0.045* |
| Phosphate | 3.20 (2.70, 3.70) | 3.70 (3.00, 4.40) | <0.0001* |
| Mg | 2.00 (1.80, 2.20) | 2.00 (1.90, 2.30) | 0.064 |
| **Inflammatory Labs** | | | |
| Ferritin | 552.70 (221.10, 1050.0) | 757.50 (313.50, 1369.0) | 0.003* |
| Albumin Serum | 3.80 (3.40, 4.10) | 3.40 (3.10, 3.70) | <0.0001* |
| Lymphocyte Count | 1.03 (0.74, 1.44) | 0.68 (0.43, 0.96) | <0.0001* |

(*Continued*)

**Table 4.** (Continued)

| Variables | Mortality Status | | |
|---|---|---|---|
| | Survival n = 801 | Non-survival n = 121 | P-value |
| Procalcitonin | 0.15 (0.09, 0.28) | 0.32 (0.16, 1.21) | <0.0001* |
| D-Dimer | 369.50 (223.00, 753.00) | 790.00 (409.00, 1941.0) | <0.0001* |
| IL6 | 42.80 (22.30, 77.10) | 87.50 (34.50, 191.80) | <0.0001* |
| WBC | 7.77 (5.77, 10.43) | 8.53 (5.53, 11.99) | 0.141 |
| ESR | 51.00 (28.00, 74.00) | 48.50 (26.00, 76.00) | 0.747 |
| CRP | 6.60 (2.35, 12.95) | 11.70 (5.70, 21.30) | <0.0001* |
| **Other Labs** | | | |
| HB | 13.20 (11.60, 14.60) | 12.60 (10.20, 14.40) | 0.038* |
| Lactate | 1.40 (1.00, 1.90) | 1.90 (1.40, 3.00) | <0.0001* |
| BNP | 156.00 (44.00, 1035.0) | 1646.0 (363.00, 4499.5) | <0.0001* |
| Troponin | 0.01 (0.01, 0.01) | 0.02 (0.01, 0.05) | <0.0001* |
| INR | 1.20 (1.10, 1.30) | 1.20 (1.10, 1.50) | 0.0002* |
| LDH | 273.00 (209.00, 364.00) | 382.00 (335.00, 546.50) | 0.0003* |
| AST | 36.00 (24.00, 58.00) | 53.00 (35.00, 76.00) | <0.0001* |
| ALT | 28.00 (17.00, 54.00) | 29.50 (16.00, 51.00) | 0.975 |
| CPK | 97.00 (50.00, 209.00) | 161.00 (61.50, 367.00) | 0.024* |
| **Lipid Profile** | | | |
| LDL | 69.00 (50.00, 90.00) | 66.00 (42.00, 81.00) | 0.276 |
| Triglyceride | 121.00 (87.00, 178.50) | 137.00 (86.00, 236.00) | 0.357 |
| HDL | 33.00 (25.00, 41.00) | 31.00 (26.00, 43.00) | 0.758 |

Data were shown with n (%) for categorical variables, mean (sd) and median (IQR) for continuous variables.

* p<0.05; P values were based on Chi-square tests, t-tests and Mann-Whitney tests.

in part, because in a multivariate analysis T1 compared T2 MABP was significantly associated with mortality in those with systemic HTN, while MABP was not statistically associated mortality in non-hypertensive subjects.

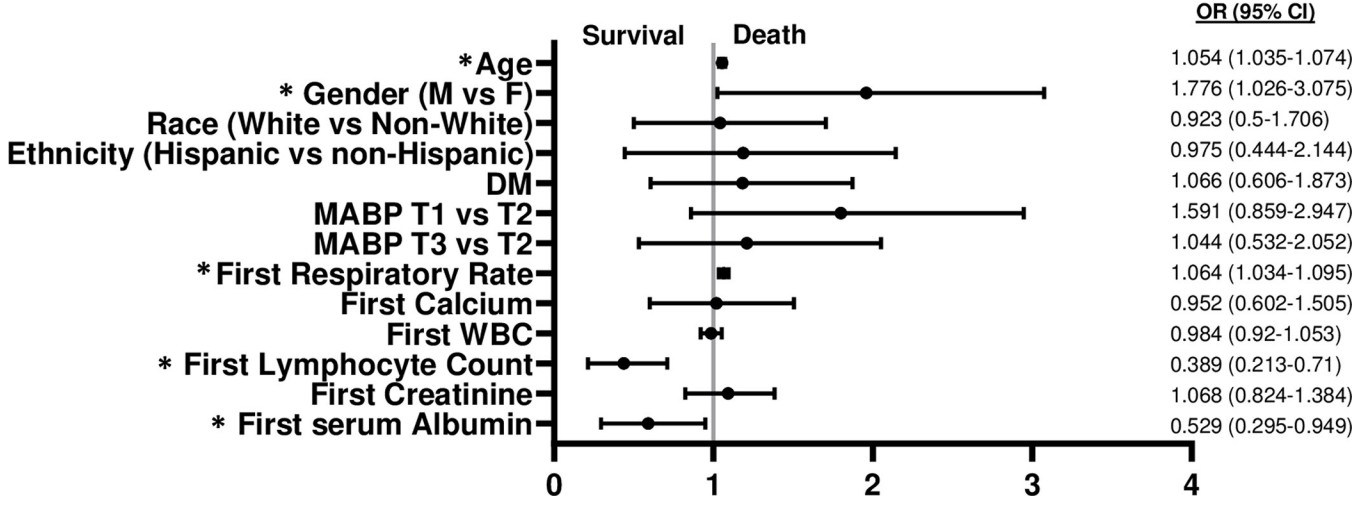

**Fig 3. The OR of variables that predict mortality in subjects without a history of HTN and hospitalized with COVID-19 (+).**

More recently, artificial intelligence (ie machine learning) models were developed to predict inpatient mortality in COVID-19 (+) patients [13]. Machine learning utilizes several types of algorithms to predict COVID-19 outcomes, and Ikemura et al showed that SBP, DBP and age were consistently high performing variables, regardless of algorithm. Moreover, these investigators showed that those patients with lower SBP had a greater risk for mortality than those with a high SBP [14]. In other words, a low and/or low-normal SBP increased your risk of death in COVID-19 (+) patients, similar to that observed in our univariate analysis of MABP [14]. In the multivariate analysis of hypertensive COVID-19 (+), the OR of death was associated with a T1 level of MABP.

This investigation has several strengths to be noted. This study included nearly 1600 COVID-19 (+) patients and ~2500 COVID-19 (-) patients hospitalized during the same period which permitted comparisons between the groups. Our COVID-19 (-) consisted of patients who had a similar presentation to the ED as those with COVID-19 and unwent similar initial management and isolation protocols until the PCR was reported (a lag time in test reporting occurred during the first wave of the pandemic). Ninety-one variables were abstracted from the medical record for each subject which included historical, clinical, and laboratory data. Moreover, only the initial wave of COVID-19 infection was analyzed because the goal was to discern the relationship between BP and SARS-CoV2, since the later waves of infection were confounded by vaccination status, natural immunity due to prior infection, SARS-CoV2 variants and new therapeutics.

However, as with all cross-sectional studies, limitations remain. Due to the observational nature of the study we cannot make any determination of casualty. The timing of the BP in relation to the PCR positive diagnosis of COVID-19 is unknown which could effect the initial BP at hospital presentation and contribute to lead-time bias in regards to mortality. An area of ongoing interest, which was not rigorously investigated, is whether ACE inhibitors or angiotensin receptors blocker effect mortality in the COVID-19 population; however, recent randomized controlled trials have not shown a moratlity benefit [15, 16]. Our univariate analysis did not show differences in mortality in hypertensive COVID-19 (+) patients taking these medications (S4 Table). Moreover, the other types of blood pressure medications administered to COVID-19 patients were not abstracted from the chart which may or may not contribute to severity of disease. Several variables did not achieve the 80% observation threshold which excluded their use in MV analysis.

In conclusion, we observe that low mean arterial BP, without overt hypotension, correlated with morality in patients with COVID-19 especially those with a history of HTN.

## Statement of ethics

The study protocol was approved by the Stony Brook University Institutional Review Board and is in compliance with the guidelines for human studies in accordance with the World Medical Association Declaration of Helsinki. Because the study design was retrospective and the data deidentified, informed consent was waived.

## Supporting information

**S1 Table. Variables measure in COVID-19(+) and COVID-19(-) hospitalized patients.**
(DOCX)

**S2 Table. Tertiles of MABP in COVID-19 (+) patients.**
(DOCX)

**S3 Table. Tertiles of MABP in COVID-19 (-) patients.**
(DOCX)

**S4 Table. Comparison of hypertensive COVID-19 positive patients who died and survived.**
(DOCX)

**S5 Table. Comparison of non-hypertensive COVID-19 positive patients who died and survived.**
(DOCX)

## Author Contributions

**Conceptualization:** Farrukh M. Koraishy, Rajeev Rohatgi.

**Formal analysis:** Tenzin Yangchen, Wei Hou.

**Investigation:** Rajeev Rohatgi.

**Methodology:** Tenzin Yangchen, Wei Hou.

**Project administration:** Rajeev Rohatgi.

**Supervision:** Farrukh M. Koraishy.

**Writing – review & editing:** Chang Xu.

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
