## [Decision Letter · Decision Letter 0]

8 Jun 2022

PONE-D-22-10325Normal mean arterial blood pressure (MABP) is a risk factor for mortality in hypertensive COVID-19 positive hospitalized patientsPLOS ONE

Dear Dr. Rohatgi,

Thank you for submitting your manuscript to PLOS ONE. After careful consideration, we feel that it has merit but does not fully meet PLOS ONE’s publication criteria as it currently stands. Therefore, we invite you to submit a revised version of the manuscript that addresses the points raised during the review process.

The authors had conducted a retrospective study to investigate the associations between MABPs and mortality in COVID-19 positive(+) patients. However, the research question is not well addressed in the manuscript. Please prepare appropriate revisions and replies to reviewers’ comments.

We look forward to receiving your revised manuscript.

Kind regards,

Yuyan Wang, Ph.D.

Academic Editor

PLOS ONE

Journal Requirements:

Additional Editor Comments:

1. Please design your analysis appropriately. Though the research question was clearly proposed in the manuscript, current results couldn’t answer the research question. Specially, if the research interest is the association between MABPs and COVID mortality, then only the COVID patients are needed. If the research interest is the association between MABPs and COVID infection, then both COVID positive and negative patients are needed.

2. Please check the distribution of the biomarkers before using mean/SD describing them. It’s obvious for some biomarkers the SDs are very large (even much larger than means). If distributions of the biomarkers are skewed, median/IQR should be used for description and nonparametric should be applied for comparison.

3. Please clarify if there are any variable selection procedure for MV analysis.

Reviewers' comments:

Reviewer's Responses to Questions

**Comments to the Author**

1. Is the manuscript technically sound, and do the data support the conclusions?

Reviewer #1: Partly

Reviewer #2: Yes

2. Has the statistical analysis been performed appropriately and rigorously? 

Reviewer #1: I Don't Know

Reviewer #2: Yes

3. Have the authors made all data underlying the findings in their manuscript fully available?

Reviewer #1: Yes

Reviewer #2: Yes

4. Is the manuscript presented in an intelligible fashion and written in standard English?

Reviewer #1: Yes

Reviewer #2: Yes

5. Review Comments to the Author

Reviewer #1: Interesting but hard to read.

First, add the meaning of abbreviation and number in

Second, in the discussion, it gives a long-winded explanation, meaning that many are on the results itself but not the interpretation and discussion on the results. And is there any evidence to support your conclusion, such as why low normal initial BP correlated with mortality and why in COVID 19 positive and why in patients with a history of HTN?

Reviewer #2: Dear authors,

Thank you for this submission.

My comments are given below:

Please add a reference or a reason for the cut-off values of MABP tertiles.

Please add some detail regarding how control group was created to the methods section. It is only given at discussion but it would be more explanatory if given at methods section as well.

Please add footnotes to the tables at supplement section to make clear meaning of asterisks.

Table 2 at amin text and Table 2 at supplement are nearly same. It is like a repetition and hard to find where the difference is. Table 2 S might be revised the form that it only contains different data from Table 2 at main text.

Please add footnote for asterisks at Table 2.

It would be well if you may add systolic, diastolic blood pressure values and mean arterial blood pressure values as well to the tables 3 and 4 instead of only giving categorizations for them.

Performing multivariate analysis is appreciated but models are created with variables significant at univariate adjustment. This made every multivariate model different from each other and avoid one to generalize results. The comment got form one figure is not valid for the other one although all were based on blood pressure. This made me confused a little bit. It is too much detail for the paper.

There would be more comparison between Covid and control cases. I would like to see if there is any difference between two groups within the same MABP tertiles. This would be speculated in some ways: highlight importance of Covid if the Covid cases’ mortality were higher or importance of blood pressure if they were similar.

6. PLOS authors have the option to publish the peer review history of their article (what does this mean?). If published, this will include your full peer review and any attached files.

Reviewer #1: No

Reviewer #2: **Yes: **Mustafa Sevinc

---

## [Author Response · Author response to Decision Letter 0]

13 Jan 2023

Response to Critique

Thank you for reviewing our manuscript and providing us an informative critique. We acknowledge the limitations noted by the reviewers and editor and hope to satisfy the critique. There has been some delay in responding to the critique because our original statistician, Ms. Yangchen, was out of the country, and after returning, started her PhD program. Dr. Wei Hou, a statistician at the Stony Brook University School of Medicine, agreed to assist in the follow up analysis. 

Reviewer#1

Interesting but hard to read.

 Thank you and will work to make the manuscript more readable.

add the meaning of abbreviation and number in

 Please see supplementary tables. This includes all the abbreviations.

in the discussion, it gives a long-winded explanation, meaning that many are on the

results itself but not the interpretation and discussion on the results. And is there any evidence

to support your conclusion, such as why low normal initial BP correlated with mortality and

(a) why in COVID 19 positive and (b) why in patients with a history of HTN

In the discussion I shortened the summary of the results to emphasize the clinically pertinent aspects and to speculate/hypothesize on mechanism of this finding. 

The study is not designed to answer why a low-normal blood pressure at initial evaluation of admitted COVID-19 (+) patients with long standing hypertension have a higher mortality. However, several clinical and basic science studies permit us to speculate why this finding may be true. Several clinical studies illustrate an association between the diagnosis of hypertension and mortality in COVID-19 (+) patients. The renin-angiotensin system, and specifically, ACE2 receptor, the receptor for COVID-19, is increased in bronchial biopsy specimens of patients with chronic systemic hypertension (1, 2). Thus, with greater availability of ACE2 receptors in hypertensive subjects, COVID-19 binding, entry and replication would be enhanced. Rapid viral replication tends to create a sicker and septic clinical picture associated with lower blood pressure; however, ACE2 receptor is a negative regulator of the renin-angiotensin system and its activation tends to reduce blood pressure. Therefore, it has been proposed that COVID-19 binding and internalization of ACE2 receptor could lead to unopposed angiontensin-2 affects leading to higher blood pressure. We asked this question about blood pressure, in part, to evaluate whether presenting blood pressure would be high (unopposed ACE2 receptor activity) or low (a more classic septic picture, with high viral load), and whether this could correlate this mortality. Based on our results, a lower MABP at presentation is a risk factor for mortality in hypertensive COVID-19 (+) subjects and, implies a septic clinical picture. The mortality in hypertensive subjects may be greater because (a) ACE2 receptors are more abundant leading to greater viral replication and entry and (b) long-standing hypertension leads to microvascular disease which, in the presence of modest hypotension, contributes to poor tissue perfusion.

Reviewer#2

1. Please add a reference or a reason for the cut-off values of MABP tertiles.

 In an earlier analysis of the data, hospitalized COVID-19 (+) patients were divided according to JNC VII guidelines. Specifically, subjects were divided into the following groups: SBP 90-140 (normotensive), SBP 141-160 (moderate hypertension) and SBP > 161 (severe hypertension). ANOVA analysis of death in these groups demonstrated statistical difference; however, when comparing death between the normotensive group, which had the lowest mortality, and moderate or severe hypertension, no significance was noted. When subjects were categorized into SBP categories, normotensive subjects were n=1112, moderate hypertensives were n=271 and severe hypertensives were n=150. We chose to divide blood pressure into tertiles of mean arterial blood pressure (MABP). Dividing into tertiles of MABP equalizes the number of subjects in each group and enhances the strengths of the conclusions. This technique has been utilized by other investigators, including the SPRINT investigators(3).

2. Please add some detail regarding how control group was created to the methods section. It is

only given at discussion but it would be more explanatory if given at methods section as well.

 All subjects (COVID-19 (-) and COVID-19 (+)) were also divided into tertiles of mean arterial blood pressure (MABP): T1 (65-85 mm Hg), T2 (86-97 mm Hg) and T3 (>98 mm Hg). For COVID-19 (+), the MABP category with the lowest mortality was identified as the control group and other two groups compared to it. The T2 MABP group had the lowest mortality amongst the three groups and was selected as the control group. It happened that in the COVID-19 (-) group that the highest mortality was in the T2 MABP group, and this was assigned as the control group.

3. Please add footnotes to the tables at supplement section to make clear meaning of asterisks.

I included the footnotes in the supplementary tables. Thank you for pointing this out.

4. Table 2 at a min text and Table 2 at supplement are nearly same. It is like a repetition and hard

to find where the difference is. Table 2S might be revised the form that it only contains

different data from Table 2 at main text.

 Thank you for this comment. I have revised the tables so that they do not repeat the same data.

5. Please add footnote for asterisks at Table 2.

 The footnote to define the asterisks has been added to the end of each table (1-4).

6. It would be well if you may add systolic, diastolic blood pressure values and mean arterial

blood pressure values as well to the tables 3 and 4 instead of only giving categorizations for

them.

 Thank you for the comment. I have added MABP values to each of the BP categories in Table 3 and 4.

7. Performing multivariate analysis is appreciated but models are created with variables

significant at univariate adjustment. This made every multivariate model different from each

other and avoid one to generalize results. The comment got form one figure is not valid for the

other one although all were based on blood pressure. This made me confused a little bit. It is

too much detail for the paper.

 Based on the Editor’s comments, the tables and MV analysis were revised. Because biomarkers are not normally distributed, all the variables were reviewed and those variables that are not normally distributed were revised to median/IQR. Statistical analyses were then performed. Based on these changes, the univariate analysis of mortality no longer was statistically significant between T3 MABP and T2 MABP (See Table 2). Therefore, MV analysis (Figure 1) only included associated risk factors evaluating T1 MABP and T2 MABP. 

8. There would be more comparison between Covid and control cases. I would like to see if there

is any difference between two groups within the same MABP tertiles. This would be

speculated in some ways: highlight importance of Covid if the Covid cases’ mortality were

higher or importance of blood pressure if they were similar.

 The mortality at each level of MABP tertile was significantly greater in the COVID-19(+) subjects than the COVID-19 (-) (p<0.0001). Though we agree that it would be interesting to compare the characteristics of the MABP tertiles between COVID-19 (+) and COVID-19 (-) patients, we feel this would require a new focus. The goal of this manuscript was to identify where the initial, presenting blood pressure predicted mortality in the COVID-19 (+). We included COVID-19 (-) data to emphasize (a) the difference in mortality between COVID-19(+) and (-) patients during the same time frame and (b) illustrate MABP’s differential associations between COVID-19 (+) and (-) subjects. Additionally, the journal editor has requested that the focus remain limited to the hypothesis and not become overly broad. 

Editor

1. Please design your analysis appropriately. Though the research question was clearly

proposed in the manuscript, current results couldn’t answer the research question. Specially, if

the research interest is the association between MABPs and COVID mortality, then only the

COVID patients are needed. If the research interest is the association between MABPs and

COVID infection, then both COVID positive and negative patients are needed.

 Thank you and we acknowledge this concern. Few COVID-19 studies reported a comparator group of uninfected patients admitted to the hospital during the initial surge of SARs-CoV-2 infections. In addition, reviewer #2 specifically stated on comparison of between MABP in both COVID-19(+) and (-) patients would be of significant interest (see above). Because of this novelty, we elected to expand the hypothesis to include uninfected subjects admitted to SBUH during the COVID surge. The hypothesis has been modified to “ Due to SARS-CoV-2’s direct effects on the RAAS system, we hypothesized that an elevated BP at clinical presentation portended greater mortality, which differed from uninfected patients.” It expands the scope of investigation, but due the observation that tertiles of MABP in COVID-19 (-) subjects were not associated with differences in mortality, no further analysis was performed.

2. Please check the distribution of the biomarkers before using mean/SD describing them. It’s

obvious for some biomarkers the SDs are very large (even much larger than means). If

distributions of the biomarkers are skewed, median/IQR should be used for description and

nonparametric should be applied for comparison.

 Thank you for this comment and the revised manuscript will follow your suggestion.

3. Please clarify if there are any variable selection procedure for MV analysis.

 In the original analysis the MV analysis was performed with variables that which statistically significant; however, after re-analysis of the data, death in T3 MABP did not differ from T2 MABP so figure 1B was removed. 

1. Pinto BGG, Oliveira AER, Singh Y, Jimenez L, Goncalves ANA, Ogava RLT, et al. ACE2 Expression Is Increased in the Lungs of Patients With Comorbidities Associated With Severe COVID-19. J Infect Dis. 2020;222(4):556-63.

2. Radzikowska U, Ding M, Tan G, Zhakparov D, Peng Y, Wawrzyniak P, et al. Distribution of ACE2, CD147, CD26, and other SARS-CoV-2 associated molecules in tissues and immune cells in health and in asthma, COPD, obesity, hypertension, and COVID-19 risk factors. Allergy. 2020;75(11):2829-45.

3. Shapiro BP, Ambrosius WT, Blackshear JL, Cushman WC, Whelton PK, Oparil S, et al. Impact of Intensive Versus Standard Blood Pressure Management by Tertiles of Blood Pressure in SPRINT (Systolic Blood Pressure Intervention Trial). Hypertension. 2018;71(6):1064-74.

---

## [Decision Letter · Decision Letter 1]

7 Mar 2023

Initial mean arterial blood pressure (MABP) is a risk factor for mortality in hypertensive COVID-19 positive hospitalized patients

PONE-D-22-10325R1

Dear Dr. Rohatgi,

We’re pleased to inform you that your manuscript has been judged scientifically suitable for publication and will be formally accepted for publication once it meets all outstanding technical requirements.

Kind regards,

Yuyan Wang, Ph.D.

Academic Editor

PLOS ONE

Additional Editor Comments (optional):

Reviewers' comments:

Reviewer's Responses to Questions

**Comments to the Author**

1. If the authors have adequately addressed your comments raised in a previous round of review and you feel that this manuscript is now acceptable for publication, you may indicate that here to bypass the “Comments to the Author” section, enter your conflict of interest statement in the “Confidential to Editor” section, and submit your "Accept" recommendation.

Reviewer #2: All comments have been addressed

2. Is the manuscript technically sound, and do the data support the conclusions?

Reviewer #2: Yes

3. Has the statistical analysis been performed appropriately and rigorously? 

Reviewer #2: Yes

4. Have the authors made all data underlying the findings in their manuscript fully available?

Reviewer #2: Yes

5. Is the manuscript presented in an intelligible fashion and written in standard English?

Reviewer #2: Yes

6. Review Comments to the Author

Reviewer #2: Dear authors,

Thank you for revised version of the manuscript.

It seems that you have answered all of my questions.

I think it may be accepted for publication.

7. PLOS authors have the option to publish the peer review history of their article (what does this mean?). If published, this will include your full peer review and any attached files.

Reviewer #2: **Yes: **Mustafa Sevinc

---

## [Editor Report · Acceptance letter]

21 Mar 2023

PONE-D-22-10325R1 

Initial mean arterial blood pressure (MABP) measurement is a risk factor for mortality in hypertensive COVID-19 positive hospitalized patients 

Dear Dr. Rohatgi:

I'm pleased to inform you that your manuscript has been deemed suitable for publication in PLOS ONE. Congratulations! Your manuscript is now with our production department. 

Kind regards, 

on behalf of

Dr. Yuyan Wang 

Academic Editor

PLOS ONE